# *Helicobacter pylori*, Vascular Risk Factors and Cognition in U.S. Older Adults

**DOI:** 10.3390/brainsci9120370

**Published:** 2019-12-12

**Authors:** Víctor M. Cárdenas, François Boller, Gustavo C. Román

**Affiliations:** 1Department of Epidemiology. Fay W. Boozman College of Public Health, University of Arkansas for Medical Sciences, Little Rock, AR 72005, USA; 2The School of Medicine & Health Sciences, The George Washington University, Washington, DC 20037, USA; 3Department of Neurology, Alzheimer & Dementia Clinic, Methodist Neurological Institute, Houston Methodist Hospital, Houston, TX 77030, USA; 4Weill Cornell Medical College, Cornell University, New York, NY 14853, USA

**Keywords:** *Helicobacter pylori*, cognition, cross-sectional studies, health surveys, aging, cobalamin, homocysteine

## Abstract

Previous studies suggested that *Helicobacter pylori* infection could be a risk factor for stroke, dementia, and Alzheimer’s disease (AD). The authors examined data from participants, 60 years old and older in the Third National Health and Nutrition Examination Survey (NHANES-III) to assess the relation between *Helicobacter pylori* infection and results of the Mini-Mental State Examination (n = 1860) using logistic regression analysis controlling for age, gender, race/ethnicity, education, poverty and history of medically diagnosed diabetes. Moreover, we examined performance on the digit-symbol substitution test (DSST) of 1031 participants in the 1999–2000 NHANES according to their *H. pylori* infection status controlling for potential confounders using multiple linear regression analyses. In 1988–1991, older adults infected with CagA strains of *H. pylori* had a 50% borderline statistically significant increased level of cognitive impairment, as measured by low Mini-Mental State Examination (MMSE) scores (age–education adjusted prevalence ratio: 1.5; 95% confidence interval: 1.0, 2.0). In 1999–2000, older US adults infected with *H. pylori* scored 2.6 fewer points in the DSST than those uninfected (mean adjusted difference: −2.6; 95% confidence interval −5.1, −0.1). The authors concluded that *H. pylori* infection might be a risk factor for cognitive decline in the elderly. They also found that low cobalamin and elevated homocysteine were associated with cognitive impairment.

## 1. Introduction

*Helicobacter pylori* infection in 1999–2000 occurred among 28% of the United States population [1]. *H. pylori* infection causes gastritis, peptic ulcer disease, non-cardia adenocarcinoma, and lymphoma of the stomach [2], as well as malabsorption of cobalamin (vitamin B_12_) due to reduced stomach acidity from *H. pylori* infection that blocks release of cobalamin from dietary protein; moreover, chronic inflammation with bacterial overgrowth increases consumption of cobalamin resulting in low serum levels [3,4]. Accumulation of homocysteine (Hcy)—an independent vascular risk factor—resulting from decreases of cobalamin and folate is associated with late-onset Alzheimer’s disease (LOAD) [5]. A number of studies have previously raised the possibility that *H. pylori* could be a risk factor for stroke and coronary artery disease [6,7,8,9,10,11,12,13,14]. A quantitative summary of seven of studies on the occurrence of *H. pylori* infection in subjects with cognitive impairment, AD and dementia indicates that there is a weak but statistically significant association between *H. pylori* infection and dementia (odds ratio OR: 1.7, 95% CI 1.1, 2.5) [15]. Two previous studies included in the review had conflicting results. Beydoun et al. [16] reported that *H. pylori*–CagA status was inversely associated with time to completion of serial digits learning test among 2438 participants ages 20–59 years old, as well as with poorer performance in recall tests among 3382 participants 60–90 years old also taking part in the Third Nutrition Examination Survey (NHANES-III). Gale et al. [17] studied 1785 subjects and found no main effects for *H. pylori* on cognitive measures among 20–59 year-old participants of the NHANES-III. Neither of the previous two studies using NHANES contributed to the primary analysis of the summary meta-analysis [15] because the outcome variable, cognitive impairment, was not dichotomous.

The purpose of our study was to test the hypothesis that *H. pylori* infection increases the prevalence of mild deficits in cognitive function using a dichotomous variable in a national sample of U.S. adults, using data of the first cycle (1988–1991) of the NHANES III and using measures of cognitive performance in the continuous scale, and the 1999–2000 cycle of the current NHANES.

## 2. Materials and Methods

### 2.1. Study Setting and Population

Details on the design and methods of NHANES have been published elsewhere [18]. In brief, the National Center for Health Statistics collects data on a representative sample of the civilian, non-institutionalized U.S. population, to monitor the nation’s health and nutritional status. By design, ethnic minorities are oversampled in NHANES to allow for evaluation of the health status of specific racial/ethnic groups and to make appropriate comparisons.

We used data from older adult (≥60 years-old) participants of the first cycle of NHANES III (1988–1991) and the 1999–2000 current NHANES who had *H. pylori* infection status as well as assessment of cognitive functions. During the first phase of the NHANES-III, a short version of the Mini-Mental State Examination (MMSE), that is, the short-portable MMSE (sp-MMSE) was used. During 1999–2000, cognitive function was assessed with the digit-symbol substitution test (DSST). We also limited the study to older adult participants with valid information on relevant covariates as follows (Figure 1):
(a)From the 33,199 records of the NHANES-III, the authors sequentially excluded 17,916 records from the second cycle of NHANES III (1992–1994) lacking data on *H. pylori* infection, 14,448 subjects <60 years of age, 792, with incomplete cognitive testing, 1050 without data on *H. pylori* infection, 66 with equivocal infection status, and the following records with missing data on poverty to income ratio (n = 310), on blood pressure measurements (n = 34), on alcohol drinking (n = 72), on self-reported medical diagnosis of diabetes mellitus (n = 2), on C-reactive protein (n = 6), on years of education (n = 10), and on sp-MMSE results (n = 15), persons with self-reported race as other/mixed (n = 30), missing data on place of birth (n = 2); after these exclusions, 1860 NHANES-III records remained in our study.(b)The authors sequentially excluded records of the 1999–2000 NHANES with missing data on DSST scores (i.e., <60 years of age *n* = 8131 and incomplete DSST tests, *n* = 417), then we excluded records of participants with missing data on *H. pylori* serology (*n* = 200), without data on blood pressure (*n* = 147), on education (*n* = 2), race/ethnicity classified as “Other/Mixed” (*n* = 20), on levels of cholesterol (*n* = 7), levels of cobalamin (*n* = 2), and place of birth (*n* = 7), leaving 1032 records from the 1999–2000 NHANES that formed the subpopulation the authors used for analysis of cognitive function using DSST scores.

### 2.2. H pylori Testing

In the first cycle of NHANES-III (1988–1991) *H. pylori* serology was assessed using a commercial IgG EIA test (Wampole Laboratories, Cranbury, NJ) that measured whole cell antibodies and specific serum antibodies to *H. pylori* cytotoxic-associated gene (CagA) protein with high sensitivity and specificity of 96% [19]. The same whole cell antibodies test for anti-*H. pylori* used in NHANES-III was employed on sera specimens of participants of the 1999–2000 NHANES, but CagA testing was not done [20].

### 2.3. Cognitive Function Testing

NHANES-III measured cognitive function of older participants using a short version of the MMSE, originally consisting of 31 questions to assess orientation, recall, attention and calculation abilities, and specifically designed to assess subjects who cannot cooperate for long periods of time [21]. The short-portable MMSE (sp-MMSE), proposed in 1975 by Pfeiffer [22] consisted of 10 items, and it deemed as cognitively impaired those with more than 3 errors. During the first cycle of NHANES-III older adults were given a 13 item sp-MMSE questionnaire assessing orientation by asking participants to provide the date of the interview and the day of the week when it took place. To assess recall the participants were told the names of three objects (“apple”, “table”, and “penny”) and then asked to repeat them immediately and later on during the interview. To assess their attention and calculation skills, participants were asked to serially subtract five times $3 from $20. In the 1999–2000 NHANES, the DSST was used. The DSST is part of the Wechsler Adult Intelligence Scale, Third Edition (WAIS-III) [23], and consists of a matrix showing pairs of digits and symbols. The participants were asked to copy symbols to match the numbers. The number of correct matches produces a score that ranged from 0 to 117 in the final study population.

### 2.4. Analysis

The entire set of 33,199 records of NHANES III, and the 9965 records of the 1999–2000 NHANES were studied to produce complex survey estimates using the subpopulation statement available in SUDAAN (version 11; Research Triangle Institute, Research Triangle Park, North Carolina). First, we described the distribution of the study population by selected variables, and those of the outcome variable (sp-MMSE) grouped as a dichotomous variable for those ranking in the lower tenth of the distribution (i.e., less than 9 out of 13 points as mild/severe cognitive impaired). This approach is consistent with other analyses of cognitive impairment among older participants of NHANES III. [24] The distribution of the DSST score was treated as the outcome variable for the 1999–2000 set of the study. The main exposure of interest, *H. pylori* infection status was treated as a dichotomous variable and three categories were formed for the 1988–1991 NHANES III according to results of *H. pylori* CagA serology: CagA positive, CagA negative and uninfected. Also, we examined as potential confounders those that previously were found to need adjustment when measuring cognitive function, according to factors such as age [15,16,17,18] and education (i.e., <high school, high school, more than high school) [25], gender, race/ethnicity [26], place of birth (U.S. vs. elsewhere), and a socioeconomic status [27], using categorical variables using the tertiles of the distribution of poverty to income ratio (PIR) [28]. The PIR was used as a proxy for socioeconomic status, and is computed as reported household income divided by the U.S. poverty threshold, which is determined annually by the Bureau of the Census and adjusted for inflation. Furthermore, we controlled the relation between *H. pylori* infection and DSST scores according to low levels of vitamin B_12_ [29], and by levels of Hcy [30] as these variables were available in the 1999–2000 NHANES dataset, but not in the first cycle of NHANES-III. Vitamin B_12_ and Hcy are known to be associated with cognitive impairment, dementia and Alzheimer’s [31,32,33].

Prevalence ratios and 95% confidence intervals (95% CI) were computed using two-by-k tables and logistic regression analysis of the NHANES-III data. The adjusted mean differences and 95% CI of DSST score were estimated using multiple linear regression analysis controlled for several potential confounders. Age, and education are well known predictors of performance on the sp-MMSE, and ethnicity and SES are known to be strongly associated with *H. pylori* infection. Therefore, we conducted separate analyses for U.S. born non-Hispanic whites, who constituted the vast majority of the NHANES-III study population according to their education level (more than high school education, high school education and less than high school education). For multiple regression analysis on DSST scores. the values of Hcy were square root transformed and used in the continuous scale as a covariate.

For variable selection we introduced all relevant covariates, and eliminated one by one those variables that had a *p*-value >0.15 and kept in the final models those variables that meaningfully changed (i.e., >10%) the value of the prevalence ratios for low sp-MMSE scores or changes of that order in the adjusted difference of mean DSST scores by *H. pylori* infection.

### 2.5. Ethical Considerations

We obtained determination of exempt of review.

## 3. Results

### 3.1. Characteristics of the Study Populations, 1988–1991 and 1999–2000

As shown in Table 1, both in 1988–1991 and in 1999–2000, older U.S. adults were typically 70 year-olds, 55% women, nearly 90% were non-Hispanic Whites and U.S.-born. In 1988–1991, 40% had less than a high-school education, ten years later this proportion decreased to 32%. Among participants with data on income, the proportion living in poverty rose slightly from 10% in 1988–1991 to 13% in 1999–2000. The proportion of older adults with high blood pressure or on high blood medications rose from 58% to 64%. The proportion with high levels of cholesterol decreased from 35% to 23%. The proportion with high levels of C-reactive protein increased from 10% to 13%, while 12% had elevated levels of Hcy only 3% had low levels of vitamin B12 in 1999–2000. The proportion with medically diagnosed diabetes remained unchanged at about 12%. In 1988–1991, the proportion of older adults infected with *H. pylori* was 53% and decreased to 38% by 1999–2000. In 1988–1991, almost half of the prevalent infections were due to CagA positive strains.

### 3.2. Cognitive Function in the Study Populations by Variables of Interest

#### 3.2.1. H. pylori and Impaired Cognitive Function in the 1988–1990 U.S. Older Adult Population

The distribution of the scores of the MMSE was negatively skewed with values ranging from 0 to 13 with a mean of 11 (SE = 0.1), a median of 12 and inter-quartile between 11 and 12. The prevalence of low (0–8/13) scores of the sp-MMSE was 9.2% (95% CI = 7.1, 11.9). As shown in Table 2, the unadjusted (i.e., crude) prevalence of mild/severe impaired cognitive function, as measured by scores less than 9/13 in the sp-MMSE, was elevated in older age, women, minorities, the poor and uneducated, hypertensive, those with chronic inflammation as measured by C-reactive protein, diabetics and those with *H. pylori* infection. Using age and education as continuous variables, the multivariate excess prevalence of mild/severe cognitive impairment among the poor and most ethnic minorities decreased significantly or completely disappeared. The excess prevalence of cognitive impairment, according to infection with CagA positive strains of *H. pylori,* decreased from an unadjusted prevalence ratio of 2.6 (95% CI: 1.7, 4.1) to a multivariate (age and education) adjusted prevalence ratio of 1.5 (95% CI: 1.0, 2.2), that is by 42%, and such excess prevalence was borderline significant at 5% level (adjusted Satterthwaite-adjusted Chi-square *p*-value = 0.12) (Panel A of Figure 2). A multivariate logistic model that did include age, education, race/ethnicity, poverty and history of medically diagnosed diabetes produced similar results regarding the differences in prevalence of cognitive impairment by *H. pylori* infection status (last column of Table 2) than the model adjusting for age and education only.

#### 3.2.2. *H. pylori* and Impaired Cognitive Function in the 1988–1990 U.S.-born non-Hispanic White Older Adult Population

The results of separate analyses among U.S.-born non-Hispanic white, older participants according to level of education are shown in Table 3. There is evidence of a stronger association between prevalence of *H. pylori* infection and prevalence of scores on the sp-MMSE less than 9/13 among participants who were more educated (i.e., prevalence ratio of 2.7 versus 1.2). The association between prevalence of cognitive impairment and *H. pylori* infection seemed stronger among U.S.-born non-Hispanic white participants with more than 12 years of education (adjusted Satterthwaite-adjusted chi-square *p*-value = 0.04).

#### 3.2.3. *H. pylori* and Cognitive Function in the 1999–2000 U.S. Older Adult Population

Turning into the analysis of the 1999–2000 NHANES, the DSST scores in the study population ranged from 0 to 117, with a mean of 46.7 (SE = 1.0); the median was 47.6 and the inter-quartile range was between 34.1 and 59.8. As shown in the Panel B of Figure 2, the weighted unadjusted mean difference in DSST scores by H. pylori infection was −9.5, but substantially decreased, by 52%, once it was adjusted by age and education to 4.6 (95% CI: −7.3, −1.8). The effects of high blood pressure and chronic inflammation as measured by C- reactive protein were less pronounced, as shown in the second model of Table 4, which controlled for 12 potential confounders. The effects of Hcy levels, low levels of vitamin B12, and medical diagnosis of diabetes were significant, and the difference of DSST scores by *H. pylori* infection was –2.6 (95% CI: −5.1, −0.1). The backward elimination of variables with the largest *p*-value meaningfully changed the previous results. Leaving a subset of these covariates in the model that included the socio-demographic factors, low-levels of B12 and Hcy, yielded a mean adjusted difference of 2.6 points in DSST scores (95% CI: −5.1, −0.1). The inclusion of elevated levels of serum creatinine, a marker of chronic kidney disease, highly correlated with Hcy and as the inclusion of both terms produced strikingly different results for these terms, the authors decided to leave in only the Hcy term. A multiple linear regression model that did not include low-levels of B_12_ and Hcy levels produced almost identical results regarding the differences in DSST scores by *H. pylori* infection status to a model without those variables.

## 4. Discussion

We found evidence of a weak association between *H. pylori* infection and cognitive impairment using both low scores on the sp-MMSE or the DSST. The scores of the sp-MMSE and DSST were adjusted by age and education, the strongest correlates of both measures, and the prevalence of mild/severe cognitive impairment decreased considerably from crude (i.e., unadjusted prevalence ratio) comparisons by *H. pylori* infection status, but remained increased and was slightly stronger among those with more education and a more homogenous life experience (U.S.-born non-Hispanic whites). The later segment comprised nearly 90% of the study participants, representing a more homogenous segment with respect to their upbringing and culture. We also found a significant deficit of 3 points in DSST scale which measures cognitive speed among older adults infected by *H. pylori* after controlling for most socio-demographic and other known risk factors for cognitive function. Such sizeable difference is equivalent to a difference produced by almost 4 years of aging.

*H. pylori* infection is a life-long infection that leads to loss of normal gastric epithelium affecting both the antrum and the corpus of the stomach resulting in decreased acid output. It has been shown in vitamin B_12_ absorption studies that *H. pylori* infection is associated with food–cobalamin malabsorption [3], which can be reversed after *H. pylori* eradication [4]. Low levels of B_12_ and high Hcy have been found to be associated in the U.S. population; elevated Hcy is an independent risk factor for dementia [31,32] and vitamin B_12_ deficiency is a well-recognized cause of dementia [5,29,30].

In addition to the above, *H. pylori* infection has been postulated as a potential factor for LOAD [34,35] affecting cognitive function by a series of mechanisms such as platelet–leukocyte aggregation, release of cytokines, eicosanoids and acute phase proteins, production of circulating lipid peroxides and reactive oxygen species that can damage endothelial barrier function, promote leukocyte adhesion and induce alterations of normal vascular function; and by increasing endothelin-1, a constrictor of arterioles and venules [34]. There is also cross mimicry between endothelial and *H. pylori* antigens, which can lead to the development of apoptosis [35,36]. However, most of these hypotheses remain untested.

It remains unclear whether the association is specific or shared with other microbes that could promote systemic chronic inflammation, as postulated elsewhere and include Herpes simplex virus type 1, *Chlamydia pneumonia*, *Borrelia burgdorferi*, prions and other infectious agents [37,38]. However, *H. pylori* is highly pathogenic and a known human carcinogen, and given the life-long duration of the infection, if untreated, and the high prevalence of *H. pylori* infection worldwide, it is probably worth a special, separate, consideration.

The sp-MMSE test used in the first cycle of the NHANES III was shorter than the version used in the second cycle, and resembles a short screening test also known as the short-portable mental status questionnaire [22], used to identify persons with at least mild cognitive impairment using a cutoff similar to the one we used in this study (≤8/13 or 62%), and in other epidemiologic studies. The questionnaire items used in the shorter version of the MMSE have not been exactly the same we used, and the cutoff values used to define the cognitive impaired group from the cognitively intact have also varied. For instance, the Iowa Established Populations for Epidemiologic Studies of the Elderly used a nine-item questionnaire and a cutoff value of 4/9 or 44%, and used a short version of a questionnaire similar to the one used in NHANES III, which identified that 2.4% of the cohort participants within 10 years of follow-up had 3–5 more errors [39]. Annweiler et al. [40] studied vitamin D intake and cognitive function among older women using sp-MMSE of ten items to define cognitive impairment. The validity of abbreviated instruments such as the sp-MMSE used in NHANES-III requires further research, but even 11-item instruments seem to capture the range of the distribution of large epidemiologic samples [41]. In addition to previous findings and methods found in the literature, we found that the empirical distribution of the sp-MMSE strongly suggested that the cutoff of <9 points out of the 13 scale is correct. On the other hand, the assessment of cognitive function in the 1999–2000 cohort was based on DSST scores of the Wechsler adult intelligence scale-revised that assesses sustained visual attention and psychomotor speed which represent cortico-subcortical functions [42]. Moreover, the entire distribution of the DSST has been found shifted by levels of blood pressure [43,44], an important risk factor for dementia, including Alzheimer’s disease [45,46].

Meta-analyses estimated that for every increase of 5 μmol/L of Hcy, the risk of coronary heart disease raises by 20%, [47] and that Hcy increases the risk of stroke as well, [48], despite negative results of randomized clinical trials using folate to decrease Hcy [49]. In addition to its damaging vascular effects, Hcy elevation has been associated with cognitive decline in the elderly, including dementia. [31] The VITACOG trial [50] demonstrated that high-dose oral B-vitamins lowered by 53% the rate of brain atrophy in subjects with Hcy > 13.0 μmol/L, in comparison with those on placebo. In 2007, Haan et al. [33] found that high Hcy was associated with greater incidence of cognitive decline and dementia (HR 2.4 95%CI 1.1, 5.2). They also confirmed that higher plasma B_12_ decreased the risk of dementia.

The interpretation of the study results, regarding a possible effect of *H. pylori* infection on impaired cognitive function requires some discussion of both its strengths and weaknesses. NHANES is a study based on a large national representative sample of the non-institutionalized U.S. population, and its measurements are of great accuracy given the measurements of high validity and reliability on several variables and health indicators. However, the assessment of cognitive function was limited to the sp-MMSE and the DSST scores and did not include other more specific tests to confirm the diagnosis of dementia, therefore the outcome variable represents the prevalence of cognitive impairment rather than dementia. Dementia is not diagnosed using psychometric tests, but clinically, by recognition of significant memory impairment and at least one of the following: agnosia, aphasia, apraxia or a disturbance in executive functioning. [51] Second, the survey data lend themselves to conduct cross-sectional studies, which cannot tell the temporal sequence of a postulated risk factor and the condition of interest (i.e., infection and impaired cognitive performance). However, *H. pylori* is acquired early in life, and therefore, occurred probably several decades before, but more importantly, some participants might have had their infection eradicated either by indication or more likely as by-stander of antibiotic use. The cross-sectional nature of the study also complicates the interpretation of the relation of DSST scores with risk factors, and indeed seriously limits the ability of the study to assess a cause-and-effect relation not only with *H. pylori* infection, but also with other risk factors of poor cognitive performance. For instance, persons at risk or already experiencing cognitive impairment or early dementia could be on B_12_ supplementation, biasing the assessment of the relation between levels of serum B_12_ and DSST scores. Iron deficiency or depletion of iron stores could mediate the association reported here, and thus require a mediation analysis of prospective, rather than cross-sectional data [52]. Third, education seems to be the most important confounder of the reported weak association between *H. pylori* infection and sp-MMSE and DDT scores, which lends to more than one interpretation. On the one hand, the cognitive scores might measure skills that are achieved by education, and therefore, education is a mediator of the association or related to socio-economic factors linked to both *H. pylori* infection and cognitive performance as measured in NHANES. Fourth, sp-MMSE and DSST scores vary by age, racial/ethnic segments, and SES as measured by education/income. The adjustment by age, and education removed most of the differences by prevalence of low sp-MMSE scores, except for differences among African-Americans, which reflects limitations of psychometric tests, which have not been calibrated to use among racial/ethnic minorities. [53,54] It has been shown in the U.S., that cognitive performance is related to racial disparities in educational attainment [55].

More research is needed to determine if the association between cognitive function and *H. pylori* holds true using different tests and in different populations. Specifically, there are opportunities to explore the potential benefits of *H. pylori* eradication therapy using well-randomized trials in high-risk populations.

## 5. Conclusions

*H. pylori* infection might be a risk factor for cognitive decline in the elderly. However, since *H. pylori* is treatable, more direct evidence is required from clinical trials.

## Figures and Tables

**Figure 1 brainsci-09-00370-f001:**
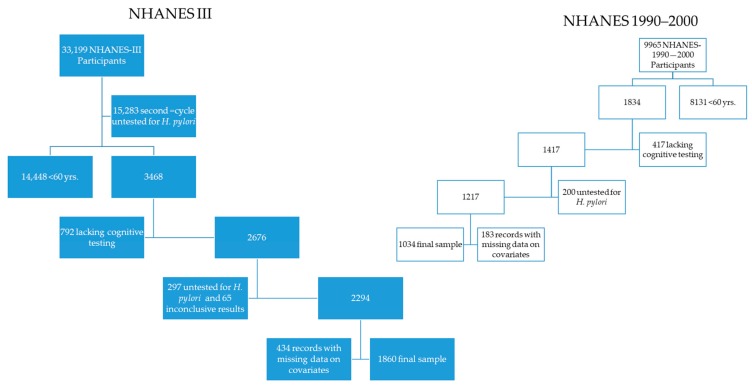
Flow chart of study populations.

**Figure 2 brainsci-09-00370-f002:**
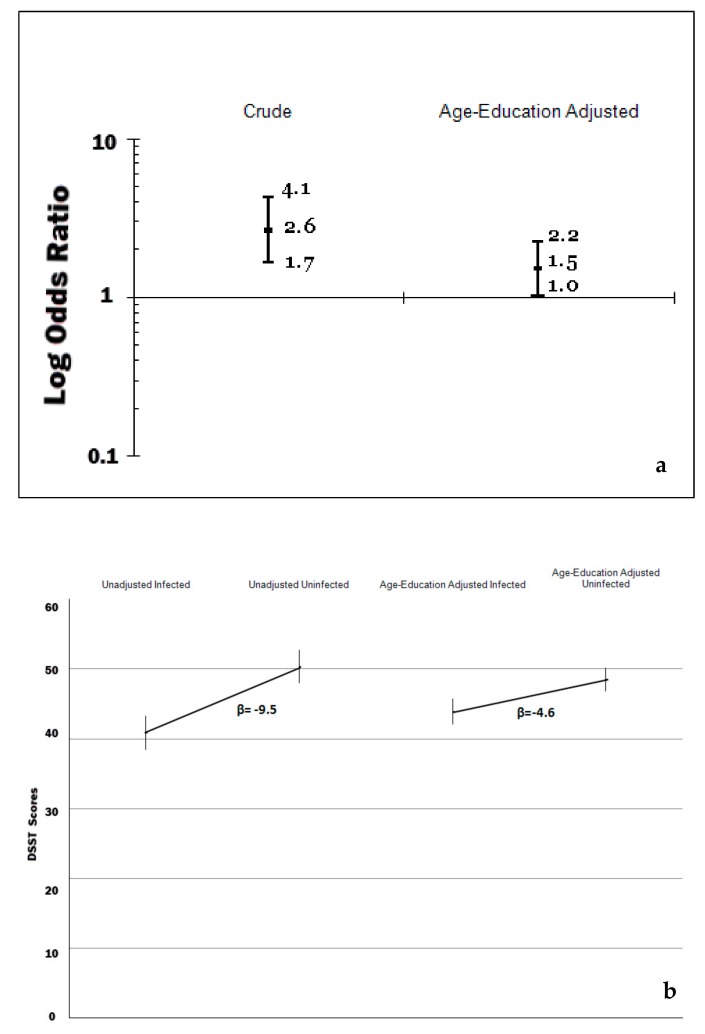
(**a**) Prevalence ratios of cognitive impairment by *H. pylori* infection among US older adults, National Health and Nutrition Examination Survey (NHANES) 1988–1991. (**b**) digit-symbol substitution test (DSST) scores among US older adults by *H. pylori* infection, NHANES 1999–2000.

**Table 1 brainsci-09-00370-t001:** Characteristics of older (60+ year-olds) adults in the study samples of the 1988–1991 National Health and Nutrition Examination Survey (NHANES) III and 1999–2000 NHANES, United States.

Characteristics	NHANES III 1988–1991		1999–2000 NHANES	
	Number	Mean/Percent (SE)	Number	Mean/Percent (SE)
**Age (years) mean**	1860	$\tilde{x}\;$= 69.9 (0.2)	1,032	$\tilde{x}\;$= 69.7 (0.3)
60–69	825	53.9 (1.7)	509	54.4 (2.1)
70–79	626	33.9 (1.4)	339	32.5 (1.6)
80+	410	12.2 (0.9)	184	13.1 (1.5)
**Sex**				
Men	952	44.4 (1.3)	522	44.7 (1.3)
Women	908	55.6	510	55.2
**Race/Ethnicity**				
Mexican American	312	1.9 (0.5)	251	2.5 (0.5)
Other Hispanic			48	5.6 (5.6)
Non-Hispanic White	1209	91.2 (1.4)	603	86.7 (2.2)
Non-Hispanic Black	339	6.9 (1.2)	130	5.2 (0.6)
**Place of Birth**				
Elsewhere	174	5.5 (1.0)	206	10.7 (2.4)
U.S.	1686	94.5	826	89.3
**Education**				
Less Than High School	1002	40.0 (3.1)	477	32.1 (3.0)
High School (or GED)	456	31.2 (1.6)	245	30.9 (2.8)
More Than High School	402	28.8 (2.6)	310	37.1 (3.3)
**Poverty to Income Ratio ***				
Under Poverty Federal Line	321	10.2 (1.4)	149	12.6 (1.8)
Above Poverty Federal Line	1539	89.8	722	87.4 (1.8)
Missing data			161	15.8 (2.7)
**Blood Pressure**				
Sys. ≥ 140 or Diast. ≥ 90 or on medications	1135	58.3 (2.4)	676	63.9 (1.9)
Absent	725	41.8	356	36.1
**Total Cholesterol**				
≥240 mg/dL	606	35.3 (1.4)	229	23.4 (1.7)
<240 mg/dL	1254	64.7 (64.7)	803	76.6
**C-reactive Protein (mg/dL)**				
<0.5 mg/dL	1360	75.0 (1.8)	684	69.6 (2.0)
0.5–0.9 mg/dL	298	14.6 (1.4)	210	17..0 (1.7)
1.0+ mg/dL	202	10.3 (1.0)	138	13.4 (1.2)
**Hyperhomocysteinemia ****				
Present			160	12.0 (1.1)
Absent			872	88.0
**Low Levels of B_12_ *****				
Present			37	2.9 (0.7)
Absent			995	97.1
**Chronic Renal Failure ******				
Present			47	3.8 (0.8)
Absent			985	96.2
**Medically Diagnosed Diabetes**				
Present	271	11.9 (1.2)	162	12.0 (1.4)
Absent	1589	88.1	870	88.0
***Helicobacter Pylori* Infection ^†^**				
Infected	1154	52.3 (1.7)	491	37.9 (2.4)
CagA positive	633	48.5 (2.5)†		
CagA negative	521	51.5 (2.5) †		
Uninfected	706	47.7 (1.7)	541	62.1

* Ratio of income to the family’s appropriate poverty threshold set by the U.S. Census Bureau. ** ≥13 μmol/L of homocysteine. *** serum B_12_ < 148 pmol/L. **** Creatinine > 137 μmol/L for men and >106 μmol/L for women. ^†^ Immune Status ratio (i.e., Optical Density ratio to corrected cutoff calibrator value) >1.10 indicates positive by IgG antibodies. The percent figures by CagA status are with respect to *H. pylori* positive by serology.

**Table 2 brainsci-09-00370-t002:** Prevalence of Mild/Severe Cognitive Impairment * among older (60+ year-olds) adults by select characteristics in 1988–1991, United States.

Characteristics	Number	Impaired (%)	Crude Prevalence Ratio (95% CI)	Age–Education Adjusted Prevalence Ratio (95% CI)	Multivariate Prevalence Ratio (95% CI)
**Age (years)**					
60–69	825	105 (6.0)	1		
70–79	626	96 (10.3)	1.9 (1.5, 2.4)		
80+	409	91 (20.1)	3.5 (2.2, 5.5)		
**Sex**					
Men	952	136 (7.7)	0.7 (0.5, 1.1)		0.7 (0.5, 1.0)
Women	908	156 (10.3)	1		1
**Race/Ethnicity**					
Mexican American	312	78 (25.7)	4.2 (2.5, 7.2)		0.8 (0.6, 1.3)
Non-Hispanic White	1209	126 (7.6)	1		1
Non-Hispanic Black	339	88 (26.2)	4.3 (3.0, 6.3)		1.9 (1.4, 2.6)
**Place of Birth**					
Elsewhere	174	34 (9.8)	1.1 (0.6, 2.0)		
U.S.	1686	258 (9.2)	1		
**Education**					
Less Than High School	1002	247 (18.7)	9.7 (5.2, 18.0)		
High School (or GED)	456	29 (3.6)	1.9 (0.7, 4.7)		
More Than High School	402	16 (1.9)	1		
Per year (0–17)			0.7 (0.7, 0.7)	0.7 (0.7, 0.8)	0.8 (0.7, 0.8)
**Poverty to Income Ratio ****					
Low tertile	856	224 (19.2)	6.6 (3.8, 11.5)		1.6 (1.0, 2.8)
Mid-tertile	554	48 (5.3)	1.8 (1.0, 3.4)		0.9 (0.5, 1.7)
Upper tertile	450	20 (2.9)	1		1
**Blood Pressure**					
Sys. ≥ 140 or Diast. ≥ 90 or on medications	1135	196 (10.5)	1.5 (1.0, 2.1)		
Absent	725	96 (7.3)	1		
**Total Cholesterol**					
≥240 mg/dL	606	92 (7.1)	0.7 (0.5,1.0)		
<240 mg/dL	1254	210 (10.3)	1		
**C-reactive Protein (mg/dL)**					
<0.5 mg/dL	1360	210 (9.0)	1		
0.5–0.9 mg/dL	298	43 (9.3)	1.0 (0.6, 1.6)		
1.0+ mg/dL	202	39 (10.5)	1.2 (0.6, 2.1)		
**Medically diagnosed diabetes**					
Present	271	66 (14.0)	1.7 (1.2, 2.4)		1.3 (0.9, 1.8)
Absent	1589	226 (8.5)	1		1
***Helicobacter pylori* Infection *****					
Infected	1154	217 (12.5)	2.2 (1.5, 3.4)	1.4 (1.0, 2.0)	1.3 (0.9, 2.0)
CagA positive	633	130 (14.7)	2.6 (1.7, 4.1)	1.5 (1.0, 2.2)	1.3 (0.9, 2.2)
CagA negative	521	87 (10.4)	1.9 (1.2, 3.0)	1.3 (0.9, 1.9)	1.2 (0.8, 1.9)
Uninfected	706	75 (5.6)	1	1	1

* <10% of the distribution of summary score of an immediate and delayed memory test, a three word registration/memory task (“apple,” “table” and “penny”) and five serial subtractions by intervals of three. ** Ratio of income to the family’s appropriate poverty threshold set by the U.S. Census Bureau. *** Immune Status ratio (i.e., Optical Density ratio to corrected cutoff calibrator value) >1.10 indicates positive by IgG antibodies.

**Table 3 brainsci-09-00370-t003:** Prevalence of Mild/Severe Cognitive Impairment* among non-Hispanic White U.S. born-older (60+ year-olds) adults by *H. pylori* infection status and level of education in 1988–1991, United States.

Education	*H. Pylori* Infected	Uninfected
	Prevalence Ratio ** (95% CI)	Referent
Less than High School	1.2 (0.7, 2.1)	1
High School	1.2 (0.5, 2.6)	1
More than High School	2.7 (0.9, 8.2)	1

* <10% of the distribution of summary score of an immediate and delayed memory test, a three word registration/memory task (“apple,” “table” and “ penny”) and five serial subtractions by intervals of three. ** Adjusted by years of age, gender, years of education within each category, poverty, and medically diagnosed diabetes.

**Table 4 brainsci-09-00370-t004:** Multivariate adjusted mean differences by select characteristics of digit-symbol substitution test scores of older adults U.S. 1999–2000.

Variables		Age–Education Adjusted		Socio-Demographic Covariates Only		All Covariates		Selected Covariates	
		Difference	(95% CI)	Difference	(95% CI)	Difference	(95% CI)	Difference	(95% CI)
*Helicobacter pylori* infection *	Present	-4.6	(−7.3, −1.8)	−2.7	(−5.2, −0.2)	−2.6	(−5.1, −0.1)	−2.6	(−5.1, −0.1)
	Absent	0	Referent	0	Referent	0	Referent	0	Referent
Age as continuous (years)		−0.8	(−0.9, −0.6)	−0.8	(−0.9, −0.6)	−0.7	(−0.9, −0.5)	−0.7	(−0.9, −0.6)
Gender	Men			−5.2	(−7.3, −3.0)	−4.4	(−6.7, −2.0)	−4.5	(−6.7, −2.3)
	Women			0	Referent	0	Referent	0	Referent
Race Ethnicity	Mexican American			−8.7	(−13.0, −4.2)	−7.8	(−12.2, −3.3)	−8.1	(−12.4, −3.8)
	Other Hispanic			−7.5	(−15.2, 0.2)	−7.4	(−14.8, 0.1)	−7.8	(−15.0, −0.7)
	Non-Hispanic White			0	Referent	0	Referent	0	Referent
	Non-Hispanic Black			−15.4	(−18.2, −12.5)	−14.4	(−17.4, −11.4)	−15.0	(−17.5, −11.7)
Education	Less Than High School	−17.1	(−20.1, −14.1)	−12.7	(−16.3, −9.1)	−12.3	(−15.8, −8.8)	−12.2	(−15.7, −8.8)
	High School (or GED)	−3.8	(−6.5, −1.1)	−3.2	(−5.8, −0.7)	−3.0	(−5.3, −0.7)	−2.9	(−5.3, −0.5)
	More than High School	0	Referent	0	Referent	0	Referent	0	Referent
Poverty Income Ratio **	Lower tertile (0–1.24)			−8.0	(−11.4 −4.6)	−7.7	(−10.9, −4.5)	−7.7	(−10.9, −4.5)
	Mid-tertile (1.25–2.90)			−4.0	(−7.3, −0.8)	−3.9	(−6.8, −0.9)	−3.7	(−6.6 −0.8)
	Upper tertile (2.93+)			0	Referent	0	Referent	0	Referent
	Missing data			−0.9	(−4.4, 2.5)	−1.1	(−4.7, 2.5)	−1.0	(−4.3,2.4)
Place of Birth	Elsewhere			−4.0	(−7.3, −0.8)	−4.3	(−9.3, 0.6)	−4.0	(−8.9, 0.9)
	U.S.			0	Referent	0	Referent	0	Referent
High Cholesterol	Present					1.0	(−1.8, 3.9)		
	Absent					0	Referent		
High Blood Pressure	Present					−1.3	(−3.2, 0.6)		
C−reactive Protein	<0.5 mg/dL					0	Referent		
	0.5–0.9 mg/dL					0.9	(−3.2, 5.0)		
	1.0+ mg/dL					−4.1	(−15.8, 7.5)		
Hyc ***						−2.3	(−4.5, −0.1)	−2.3	(−4.4, −0.2)
Low levels of B_12_ ****	Present					−4.7	(−9.5, 0.1)	−4.7	(−9.2, −0.2)
	Absent					0	Referent	0	Referent
Medically diagnosed diabetes	Present					−3.9	(−7.6, −0.1)	−4.1	(−7.8, −0.4)
	Absent					0	Referent	0	Referent

* Immune Status ratio (i.e., Optical Density ratio to corrected cutoff calibrator value) >1.10 indicates positive by IgG antibodies. ** Ratio of income to the family’s appropriate poverty threshold set by the U.S. Census Bureau. *** square root transformed **** serum B_12_ < 148 pmol/L.

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
