# Peer review of "Helicobacter pylori, Vascular Risk Factors and Cognition in U.S. Older Adults"

_brainsci, 2019, doi:10.3390/brainsci9120370_

Round 1
Reviewer 1 Report
This is a well-written manuscript addressing an important topic on relationship between H. pylori infection and cognitive function in adults in the USA using national general population samples.
H. pylori was linked to diminished iron stores (e.g., iron deficiency anemia), which might affect cognitive abilities and the risk of dementia. Did the authors address such associations in their datasets? A comment on this point in the discussion is warranted.
Material and methods section
Please provide information, if available, on the validity of the MMSE cutoff value that was used to define impaired cognitive function.
Lines 72-86: Flow charts describing the selection of the study sample can be very helpful.
Lines 88-93: H. pylori testing: please clarify whether CagA was tested just in H. pylori positive samples or in all samples.
Statistical methods: Lines 121-122: "education" appears twice.
Tables 2/3: Please add a footnote regarding the definition of cognitive impairment.
Author Response
Reviewer 1:
General Comment:
“This is a well-written manuscript addressing an important topic on relationship between H. pylori infection and cognitive function in adults in the USA using national general population samples.”
Reply: We were pleased with the reviewer finding our study and paper addressing an important topic, and finding the manuscript well written. We are addressing each of the specific comments below:
“H. pylori was linked to diminished iron stores (e.g., iron deficiency anemia), which might affect cognitive abilities and the risk of dementia. Did the authors address such associations in their datasets? A comment on this point in the discussion is warranted.”
Reply: We agree that iron-deficiency and depletion of iron stores might be at work in the association reported here. We think however, that the best way to address the potential mediation of the association by nutrient deficiencies would be prospective data, from observational studies, or even better from experimental studies as we had proposed the NIH to fund. In the absence of such data, we added the following change to the discussion: (Line 313) “Iron deficiency or depletion of iron stores could mediate the association reported here, and thus require a mediation analysis of prospective, rather than cross-sectional data [49].”
“Material and methods section
Please provide information, if available, on the validity of the MMSE cutoff value that was used to define impaired cognitive function.
Reply: We have used an approach similar to that described in a paper by Noble JM and described in our text as a lower 10% of the distribution, and we acknowledge in the discussion section, that the measurements of cognitive functions have limited validity.
Lines 72-86: Flow charts describing the selection of the study sample can be very helpful.
Reply: we have added a flow chart below the text describing the selection of the study sample.
Lines 88-93: H. pylori testing: please clarify whether CagA was tested just in H. pylori positive samples or in all samples.
Reply: Cag was tested in all samples. The text reads as follows: “that measured whole cell antibodies and specific serum antibodies to H. pylori cytotoxic-associated gene (CagA) protein with high sensitivity and specificity of 96% [19].”
I ran the analysis for reassurance of the reviewer:
|
The SAS System |
The FREQ Procedure
|
|
||||||||||||||||||||||||||||||||||||||||||||||||||||||||||
Statistical methods: Lines 121-122: "education" appears twice.
Reply: thanks for pointing my mistake.
Tables 2/3: Please add a footnote regarding the definition of cognitive impairment.
Reply: Footnotes added, thanks.
Reviewer 2 Report
The paper entitled: Helicobacter pylori, vascular risk factors and cognition in US older adults by Víctor M. Cárdenas et al tests the hypothesis that H. pylori infection increases the prevalence of mild deficits in cognitive function, using large data of the first cycle (1988-1991) of the NHANES III and the 1999-2000 cycle of the current NHANES.
The methods and results are well done, the tables and figures are clear but the introduction and the discussion are poor.
The authors did not discuss the role of other Microbes and gut microbiota on Alzheimer' disease, dementia and deficits in cognitive function.
Recent animal, human, epidemiological and in-silico studies, showed a role of H. pylori and/or dysbiosis in Alzheimer' disease, and dementia mostly through the promotion of systemic chronic inflammation and/or by triggering molecular mimicry mechanisms.
Moreover other bacteria not only H pylori for example: Clamidya, CMv, HSV have been shown to affect neurodegeneration by promoting inflammation, inducing molecular mimicry mechanisms and accumulation of Amiloid β into the brain.
Other recent theories suggest the possibility of the bacterium to enter in contact with the brain through the oral-, nasal-, olfactory-pathway or through the disrupted blood-brain barrier, thus triggering the process of neurodegeneration.
All these issues should be mentioned and discussed to improve the quality of the article.
Minor change:
H.pylori has to be written in the same way and in italics
The references has to be written with the same character and in the same way
Author Response
Reply to Reviewer 2:
General Comment:
“The methods and results are well done, the tables and figures are clear but the introduction and the discussion are poor.”
Reply: We were pleased with the reviewer finding our study methods and presentations of the results well written. We are addressing each of the specific comments below:
The authors did not discuss the role of other Microbes and gut microbiota on Alzheimer' disease, dementia and deficits in cognitive function. Recent animal, human, epidemiological and in-silico studies, showed a role of H. pylori and/or dysbiosis in Alzheimer' disease, and dementia mostly through the promotion of systemic chronic inflammation and/or by triggering molecular mimicry mechanisms. Moreover other bacteria not only H pylori for example: C(h)lamidya, CMv, HSV have been shown to affect neurodegeneration by promoting inflammation, inducing molecular mimicry mechanisms and accumulation of Amiloid β into the brain.
Other recent theories suggest the possibility of the bacterium to enter in contact with the brain through the oral-, nasal-, olfactory-pathway or through the disrupted blood-brain barrier, thus triggering the process of neurodegeneration.
All these issues should be mentioned and discussed to improve the quality of the article.
Reply: We have added the following to the discussion and refer the readers to the review quoted by the reviewer: “It remains unclear whether the association is specific or shared with other microbes that could promote systemic chronic inflammation, as postulated elsewhere and include Herpes simplex virus type 1, Chlamydia pneumonia, Borrelia burgdorferi, prions and other infectious agents (37, 38). However, H. pylori is highly pathogenic and a known human carcinogen, and given the life-long duration of the infection, if untreated, and the high prevalence of H. pylori infection worldwide, it is probably worth a special, separate, consideration.”
Minor change:
H.pylori has to be written in the same way and in italics
Reply: Thanks!
The references has to be written with the same character and in the same way
Reply: Thanks!
Round 2
Reviewer 2 Report
The authors replied correctly and improved the quality of paper that can be published